# Evaluating the Effectiveness of Letter and Telephone Reminders in Promoting the Use of Specific Health Guidance in an At-Risk Population for Metabolic Syndrome in Japan: A Randomized Controlled Trial

**DOI:** 10.3390/ijerph20053784

**Published:** 2023-02-21

**Authors:** Hiroshi Murayama, Setaro Shimada, Kosuke Morito, Haruna Maeda, Yuta Takahashi

**Affiliations:** 1Research Team for Social Participation and Community Health, Tokyo Metropolitan Institute of Gerontology, Tokyo 173-0015, Japan; 2Health and Welfare Bureau, Yokohama 231-0005, Japan

**Keywords:** specific health guidance, reminder, randomized controlled trial, Japan

## Abstract

Japan has introduced a nationwide lifestyle intervention program (specific health guidance) for people aged 40–74 years. Medical insurers apply a reminder system to improve their utilization rates. This study examined the effectiveness of two methods of reminders (mailed letters and telephone calls) in a randomized controlled trial. Subscribers to National Health Insurance in Yokohama City, Kanagawa Prefecture, who were eligible for specific health guidance in 2021, were recruited. A total of 1377 people who met the criteria of having or being at risk of developing metabolic syndrome (male: 77.9%, mean age: 63.1 ± 10.0 years) were randomly assigned to one of three groups: a “no reminder” group, a “letter reminder” group, or a “telephone reminder” group. The utilization rates of specific health guidance were not significantly different between the three groups (10.5%, 15.3%, and 13.7%, respectively). However, in the case of the telephone reminder group, a subgroup analysis showed that the utilization rate was significantly higher among participants who received the reminder than those who did not answer the calls. Although the effectiveness of a telephone reminder might be underestimated, this study suggests that neither method impacted the utilization rates of specific health guidance among the population at risk of metabolic syndrome.

## 1. Introduction

Cardiometabolic diseases (CMDs), such as cardiovascular disease, stroke, diabetes, and chronic kidney disease, are the leading causes of mortality worldwide [1]. The main cause of CMDs is a cluster of metabolic derangements known as metabolic syndrome. The underlying factors for the incidence of metabolic syndrome include obesity, physical inactivity, and older age [2]. Therefore, with the increasing rates of obesity [3] and adoption of sedentary lifestyles [4], in combination with the aging of the global population, there is an urgent need to screen for CMD risks and to derive novel prevention strategies. In this context, several countries, including Japan, are striving to establish screening programs and lifestyle intervention strategies in order to promote the primary prevention of CMDs.

In 2008, Japan introduced a nationwide screening program (i.e., health checkups) to identify individuals with high obesity and cardiovascular risks (known as metabolic syndrome). In addition, the country established specific health guidance (i.e., a lifestyle intervention program) to reduce cardiovascular risk factors [5]. These services are provided to all adults aged 40–74 years every year and are delegated to medical insurers by the Act on Securing Medical Care for the Elderly. In fact, medical insurers have taken various measures to improve the utilization rates of the programs by the targeted population. According to recent statistics collected in Japan, approximately 30 million people underwent screening and 1.2 million people utilized specific health guidance in 2019 [6].

Meta-analyses revealed that lifestyle intervention can reduce cardiometabolic risks [7,8,9,10]. In contrast, a large-scale community-based study concluded that individually tailored lifestyle interventions had no effect on ischemic heart disease, stroke, or mortality on the population level after 10 years [11]. In addition, Japanese studies using a quasi-experimental design reported limited effects of lifestyle interventions on cardiometabolic risk factors [12]. Thus, evidence for the effects of health guidance remains controversial. Nevertheless, since specific health guidance is already a national measure, considering better methods to improve the utilization rate is important for both current and future systems.

Many medical insurers in Japan use reminders to improve the utilization rates of specific health guidance. Systematic reviews have reported the effectiveness of such reminder systems for cancer screening [13]. In the case of general health checkups, a study in the United Kingdom showed that the utilization rate was higher when short message services were used as reminders compared to when the usual letter reminders were used [14]. It has also been reported that telephone reminders are more effective in increasing participation in health checkups than letter reminders [15]. However, the effectiveness of reminders seems to differ depending on the population’s demographic characteristics, such as race/ethnicity [16]. Previous findings mainly originate from studies conducted in Western countries. Therefore, it is necessary to confirm whether these findings are applicable to other populations, such as those in Japan.

The purpose of this study was to examine the effectiveness of reminders in promoting the utilization of specific health guidance using a randomized controlled trial design. In this study, we used two reminder methods (letters and telephone calls). In addition, this study focused on people who are considered at high risk of metabolic syndrome, given the requirement for reminders among this population.

## 2. Materials and Methods

### 2.1. Sample and Procedures

The target population was National Health Insurance subscribers in Yokohama City, Kanagawa Prefecture, Japan (approximately 510 thousand subscribers as of April 2021). Yokohama is the capital of Kanagawa Prefecture and is located 30 km southwest of central Tokyo. As of April 2021, Yokohama City had a population of approximately 3.78 million. At the time of this study, National Health Insurance covered approximately 20% of the total city population.

Among the National Health Insurance subscribers in Yokohama, 460,928 were eligible for health checkups in the fiscal year (FY) of 2021 (i.e., between April 2021 and March 2022), and 113,945 received health checkups. Of these, 13,638 were eligible for specific health guidance based on national criteria. Among them, 10,763 people who were deemed to require immediate medical attention (based on the national criteria using the results of health checkups, renal function tests, blood pressure, complete blood counts, lipid panels, blood sugar levels, and liver function tests) were excluded. Consequently, we included 1377 people (355 met the criteria for metabolic syndrome and 1022 were considered to be at risk of metabolic syndrome). In FY2021, among those who underwent health checkups, 14.8% and 56.9% were judged as “applicable” and “at risk” of metabolic syndrome, respectively. The implementation rate of specific health guidance in Yokohama City was 9.3% in FY2020.

This study adopted a random sampling method to enhance the internal validity of the findings. The participants were randomly assigned to three groups: the “no reminder” group (n = 458), the “letter reminder” group (n = 459), or the “telephone reminder” group (n = 460). Random assignment was conducted by the staff of Yokohama City. The staff provided a unique number to every participant and assigned them randomly to one of the three groups using a random number generator. This process was performed each month. The data analysts, but not the participants, were blinded to the information on the group assignment. A flow diagram of the sampling and allocation processes is shown in Figure 1.

### 2.2. Intervention

We adopted letter and telephone reminder interventions in this study. The interventions were administered by the staff of Yokohama City. The information provided to the participants via either letter or telephone call was not personalized.

#### 2.2.1. Letter Reminder

A reminder was mailed to the participants’ home addresses. The main components of the letter were an “explanation of the specific health guidance (including information that the specific health guidance was free of charge)”, “the expiration date of the specific health guidance”, “information on the medical centers/hospitals/clinics where the specific health guidance is provided”, and “telephone number for inquiries”. The expiration date was determined according to the month in which the participants underwent a health checkup. The coupon for specific health guidance was valid for two months from the time of dispatch.

#### 2.2.2. Telephone Reminder

The public health nurse called the participants on weekdays using the phone numbers that the participants had provided as their contact information when they were enrolled in the National Health Insurance program. The information provided to the participants was compiled into a manual. The main contents were “a brief explanation of the results of the health checkups”, “the explanation of the specific health guidance (including information that the specific health guidance was free of charge)”, “the expiration date of the specific health guidance”, and “information on the medical centers/hospitals/clinics where the specific health guidance is provided and the way to make an appointment”. In cases of disconnection, the public health nurse re-called the participant on different weekdays (up to three times). If family members answered the phone, the public health nurse told them to re-call on different days and asked them to encourage the participant to receive specific health guidance.

Of 460 individuals assigned to the telephone reminder group, the public health nurse was able to directly reach 274 participants (59.6%) and to leave a message with the family members of 34 participants (7.4%). The public health nurse could not reach the remaining 152 participants, and they did not receive the telephone reminder.

### 2.3. Measures

#### 2.3.1. Outcomes

The outcome variable was whether or not the participants utilized specific health guidance in FY2021. Information on the participants’ use of specific health guidance was obtained from the Data Management System of Yokohama City.

#### 2.3.2. Participants’ Characteristics

We used the participants’ demographics (sex and age) and the results of the health checkups obtained via the Data Management System. The results of the health checkups included the abdominal circumference, body mass index, diastolic blood pressure, systolic blood pressure, HbA1c, fasting blood glucose, triglyceride, high-density lipoprotein cholesterol, history of diseases (cerebrovascular diseases, cardiovascular diseases, chronic kidney failure, and dialysis therapy), smoking habits (“Do you currently smoke habitually?”; yes or no), exercise habits (“Do you exercise lightly for at least 30 min two days a week for at least one year?”; yes or no), and frequency of drinking (“How often do you drink alcohol?”; every day, sometimes, rarely, or never).

### 2.4. Statistical Analysis

First, the participants’ characteristics were compared between the three groups using the chi-square test, Fisher’s exact test, and Kruskal–Wallis test. For continuous variables, confirmed to be not normally distributed by the Shapiro–Wilk test, the non-parametric test was conducted (i.e., the Kruskal–Wallis test). Second, the outcome variable (i.e., the utilization of specific health guidance) was compared between the three groups using the chi-square test. For multiple comparisons, the Bonferroni correction was adopted with a significance level (α) of 1.7% (i.e., *p* < 0.017 (=0.05/3)). The analysis was performed using IBM SPSS Statistics 29 (IBM Corp., Armonk, NY, USA).

Previous studies have suggested that reminders of health checkups increase the uptake rate by a factor of approximately 1.5–1.7 times [14]. The utilization rate of specific health guidance in Yokohama City in FY2020 was 9.3% in total. However, it was 13.3% for the subpopulation of people who met the criteria for metabolic syndrome or were at risk of metabolic syndrome. Assuming a significance level (α) of 0.05 (actually calculated as 0.017, considering multiple comparisons) and a power (1 − β) of 0.80, the total number of necessary samples was projected to be 1380 cases (460 in each group).

## 3. Results

Table 1 shows the characteristics of the participants. Overall, 77.9% were male, and the average age was 63.1 ± 10.0 years. No differences were observed between the three groups in any of the variables.

Table 2 presents the utilization rates of specific health guidance among the three groups. The utilization rates were 10.5% in the no-reminder group, 13.7% in the letter reminder group, and 15.3% in the telephone reminder group, with no significant differences between the three groups (χ^2^ = 4.753, *p* = 0.093). Moreover, no differences were found between any two groups in multiple comparisons.

Although they are not shown in the table, we compared the utilization rates of the participants whose calls were answered either by them directly or by their family members (n = 308, 67.0%) and those who were not reachable by telephone (n = 152, 33.0%) in the telephone reminder group. The utilization rates were 16.9% (52 of 308) and 7.2% (11 of 152), respectively (χ^2^ = 8.012, *p* = 0.004). This difference remained significant even after adjusting for sex, age, body mass index, and history of disease.

## 4. Discussion

In this study, we examined the effectiveness of two reminder methods in regard to the rate of utilization of specific health guidance (i.e., letters and telephone calls) using a randomized controlled trial design. Most medical insurers in Japan use a call–recall methodology to improve the implementation rate of specific health guidance. However, its effectiveness has not been yet sufficiently verified. This study focused on widely used reminder methods that can contribute to the establishment of evidence-based health activities.

The analysis did not demonstrate an improvement in the utilization rate after either letter or telephone reminders compared to no reminder. This result differs from those of previous studies regarding general health checkups and cancer screening [13,14,15,17], which confirmed the effectiveness of reminders. One possible reason for this inconsistency may be that individuals refrained from following specific health guidance due to the recent coronavirus disease 2019 (COVID-19) pandemic. The COVID-19 pandemic has affected many aspects of people’s behaviors in daily life. In fact, the nationwide implementation rate of specific health guidance had been increasing every year until FY2019 (i.e., before the COVID-19 outbreak); however, in FY 2020, during the outbreak, it decreased (overall: from 29.3% to 27.9%, male: from 27.5% to 26.4%, and female: from 32.9% to 30.9%) [6]. The spread of COVID-19 has augmented people’s fear of going out and visiting places where people gather, and thus the utilization of specific health guidance might be impacted. Another possibility is the limited population targeted in this study, i.e., people who have metabolic syndrome or are at risk of developing metabolic syndrome. Although we focused on this population because of their higher need for lifestyle interventions, they might have special circumstances that hinder their use of specific health guidance, which could lead to an underestimation of the effectiveness of reminders. We considered these two possibilities in our analysis.

Earlier studies regarding the use of a letter invitation/reminder to attend health checkups reported a lack of impact of letters [16,18,19]; however, the letter is the most common method used to invite/remind individuals about health checkup participation. This study also revealed that the effect of the letter reminder might be weak. The letter reminder used in this study was generic. Since some studies demonstrated that tailoring messages in the letter to the individual’s level of risk could increase the participation rate in cancer screening programs [20,21], personalization of the letter may lead to a higher level of utilization of health guidance.

Previous studies revealed that telephone reminders are more effective than letter reminders [15,17]. A qualitative study conducted in the United Kingdom reported that participants could directly make an appointment for consultation or to obtain health guidance via telephone reminder, which could contribute to increased utilization [22]. However, in Yokohama City, owing to the system of specific health guidance, it was not possible to make an appointment for specific health guidance over the phone directly; the participants had to make an appointment later by themselves. The inefficiency of this process may have reduced the effectiveness of telephone reminders. In addition, the telephone reminder group included both participants who could be reached by a public health nurse and those who could not; thus, the effectiveness of the telephone reminder could have been underestimated. However, as specific health guidance is provided for those aged 40–74 years, including the working-age population (e.g., those in their 40s and 50s), from a practical perspective, it is difficult to access all of the target population when the telephone reminder is performed on weekdays. To increase the effectiveness of telephone reminders, a more flexible system, such as calling in the evening/nighttime or on weekends, should be implemented.

Although this study showed no difference in the utilization rate of specific health guidance between the three groups, this does not necessarily mean that reminders are ineffective. This study focused on individuals who met the criteria for metabolic syndrome or were considered at risk of developing metabolic syndrome. Therefore, the findings suggest that reminders directed towards this population may be given lower priority. Based on this study, future investigations could be conducted to verify which populations will benefit most from the reminders system.

This study has several limitations. First, as previously mentioned, the current study targeted only those with metabolic syndrome or subjects at risk of developing this disease. The effects on other populations need to be investigated in future studies to enhance the external validity. Second, this study was conducted in Yokohama. The possibility that the results may differ between regions with different medical resources and resident characteristics cannot be denied, and the generalizability of the findings must be carefully considered. Third, this study was performed in FY2021, the second year of the COVID-19 pandemic. People’s attitudes toward the utilization of specific health guidance could have been influenced by the outbreak. Therefore, the present findings might not necessarily be applicable to the “post-COVID-19 era”. Fourth, many other factors prevent people from using specific health guidance (e.g., the inconvenience of making an appointment for specific health guidance and inaccessibility of the implementation site). Therefore, it might not be sufficient to improve the utilization rate by implementing reminders alone. Fifth, we were not able to investigate how many participants in the letter reminder group actually read the letter. The effects of the letter reminder might have differed between those who read the letter and those who did not. This means that we might have underestimated the effectiveness of the letter. Finally, this study investigated the effectiveness of letter and telephone reminders. However, there are some other reminder options (e.g., short message service (SMS) and e-mail), and these effects should be examined in the future.

## 5. Conclusions

We examined the effectiveness of two types of reminder methods (i.e., letters and telephone calls) in regard to the utilization of specific health guidance using a randomized controlled trial design for individuals with metabolic syndrome or those who were at risk of developing it. The results suggest that low priority is assigned to the task of reminding people in the population at risk of metabolic syndrome. Nonetheless, this study possibly underestimated the effectiveness of reminders. Reminders using either letters or telephone calls are labor- and cost-intensive to some degree. Thus, more effective and efficient methods should be explored for the implementation of reminders.

Medical insurers utilize a reminder method to increase the implementation rate of health checkups and health guidance worldwide. Previous studies regarding the effectiveness of reminders are mainly derived from Western countries such as the United Kingdom. However, as it has been implied that the population’s demographic characteristics may affect the effectiveness of reminders [16], a study should be conducted to clarify the effectiveness of the reminder methods in each context. In addition, there are other methods, such as SMS and e-mail, other than the letter and telephone reminder methods investigated in this study, and more effective methods will probably emerge with the advancement of technology. Such methods also need to be verified using a robust design.

## Figures and Tables

**Figure 1 ijerph-20-03784-f001:**
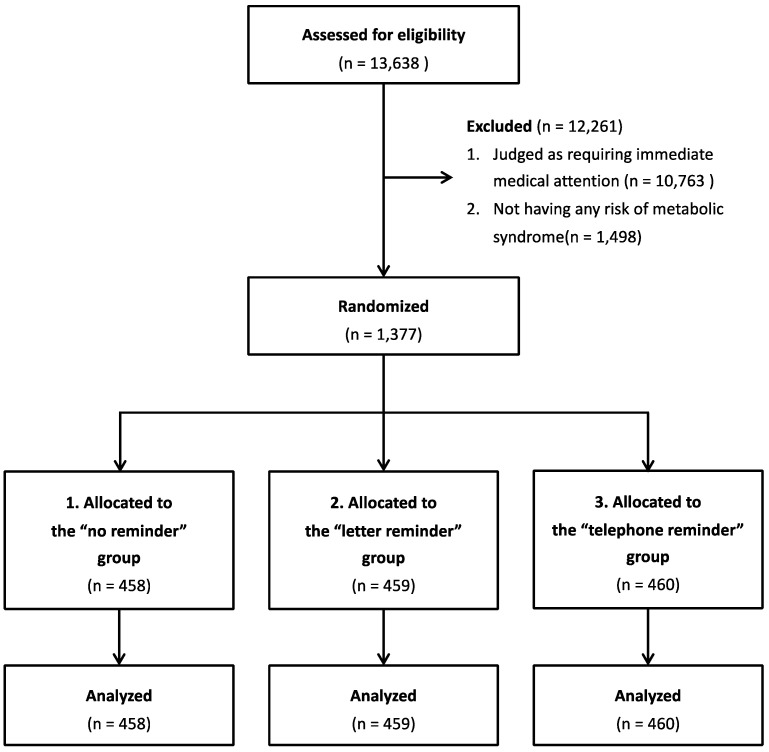
Flow diagram of the sampling and allocation processes.

**Table 1 ijerph-20-03784-t001:** Participants’ characteristics.

Variable	Category	Total n = 1377	1. No Reminder n = 458	2. Letter Reminder n = 459	3. Telephone Reminder n = 460	*p*
Sex	Men	1073 (77.9)	357 (77.9)	357 (77.8)	359 (78.0)	0.995 ^a^
Age (years old)		63.1 ± 10.0	63.3 ± 9.8	63.0 ± 10.0	62.9 ± 10.1	0.857 ^c^
Waist circumference (cm)		92.3 ± 6.0	92.3 ± 5.8	92.1 ± 5.6	92.6 ± 6.5	0.841 ^c^
Body mass index (kg/m^2^)		25.5 ± 2.8	25.5 ± 2.6	25.4 ± 2.7	25.6 ± 2.9	0.721 ^c^
Diastolic blood pressure (mmHg)		126.7 ± 9.8	126.5 ± 10.1	127.4 ± 9.2	126.3 ± 10.1	0.549 ^c^
Systolic blood pressure (mmHg)		76.5 ± 8.3	75.9 ± 9.1	76.8 ± 7.9	76.7 ± 7.9	0.473 ^c^
HbA1c (%)		5.5 ± 0.3	5.5 ± 0.3	5.5 ± 0.4	5.5 ± 0.3	0.409 ^c^
Fasting blood glucose (mg/dl)		97.7 ± 10.7	97.4 ± 10.3	98.2 ± 11.2	97.4 ± 10.6	0.753 ^c^
Triglyceride (mg/dl)		134.5 ± 62.1	136.6 ± 62.0	131.0 ± 63.2	135.8 ± 61.2	0.290 ^c^
High-density lipoprotein cholesterol (mg/dl)		57.7 ± 15.9	57.9 ± 15.8	58.0 ± 16.5	57.1 ± 15.3	0.766 ^c^
History of diseases	Cerebrovascular diseases	14 (1.0)	2 (0.4)	7 (1.5)	5 (1.1)	0.103 ^b^
	Cardiovascular diseases	51 (3.7)	17 (3.7)	12 (2.6)	22 (4.8)	0.220 ^a^
	Chronic kidney failure or under dialysis therapy	2 (0.1)	1 (0.2)	0 (0.0)	1 (0.2)	0.606 ^b^
Smoking habits	Having	267 (19.4)	82 (17.9)	82 (17.9)	103 (22.4)	0.137 ^a^
Exercise habits	Having	584 (42.4)	207 (45.2)	179 (39.0)	198 (43.0)	0.156 ^a^
Frequency of drinking	Every day	448 (32.5)	154 (33.5)	154 (33.6)	140 (30.4)	0.390 ^c^
	Sometimes	394 (28.6)	132 (28.8)	118 (25.7)	144 (31.3)	
	Rarely/never	535 (38.9)	172 (37.6)	187 (40.7)	176 (38.3)	

Values represent n (%) or mean ± standard deviation. ^a^: Chi-square test, ^b^: Fisher’s exact test, ^c^: Kruskal–Wallis test.

**Table 2 ijerph-20-03784-t002:** Comparison of the utilization rate of specific health guidance between the groups.

Variable	Category	1. No Reminder n = 458	2. Letter Reminder n = 459	3. Telephone Reminder n = 460	*p*	
					Among Three Groups	Multiple Comparison ^a^
Utilization of specific health guidance	Yes	48 (10.5)	70 (15.3)	63 (13.7)	0.093 (χ^2^ = 4.753)	1 vs. 2: 0.038 1 vs. 3: 0.156 2 vs. 3: 0.513
	No	410 (89.5)	389 (84.7)	397 (86.3)		

Values represent n (%). ^a^: The significance level was set as 0.017 (=0.05/3).

## Data Availability

The datasets used and analyzed during the current study are available from the corresponding author upon reasonable request.

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
