# Peer review of "Evaluating the Effectiveness of Letter and Telephone Reminders in Promoting the Use of Specific Health Guidance in an At-Risk Population for Metabolic Syndrome in Japan: A Randomized Controlled Trial"

_ijerph, 2023, doi:10.3390/ijerph20053784_

Round 1
Reviewer 1 Report
Evaluating the effectiveness of letter and telephone reminders in promoting use of specific health guidance in an at-risk population with metabolic syndrome in Japan: A randomized controlled trial - ijerph-2169321-peer-review-v1
While the paper describes the sampling method at length, crucial details are missing. For example, how and why were the participants randomly assigned to three groups each month: the "no reminder” group (n = 458), the "letter reminder” group (n = 459), or the "telephone reminder” group (n = 460)? How was the survey ‘administered’? What are the assumptions for use of the chi-square test, Fisher's exact test, Kruskal-Wallis test, Bonferroni correction and one-way analysis of variance? Was the data set tested to assess its compliance with these assumptions? How did you handle the issue of hidden and overt bias? Can you attribute the changes recorded in the outcome to the telephone and/or letter reminder? Of course there are other factors (hidden and unhidden) that could course the changes. How did you handle these hidden and unhidden factors methodologically and econometrically?
The discussion of the results needs to be improved. Currently, it is unfocused. In addition, much of the discussion is reiterating the results, while there is only limited discussion of the findings in light of previous and related literature. The discussion section overall would be improved by a much clearer focus on elucidating key findings in light of existing literature and how this study builds on/relates to past findings. Specifically, there needs to be significantly more references and linkage to relevant literature to back up many of the points made in this section.
This conclusion is poorly written and needs to be reworked. Look at this excerpt from the section – “There was no difference in the utilization rate for either letter or telephone reminders compared with no reminders. However, in the telephone reminder group, 33.0% of the participants could not be reached through phone calls, and their utilization rate was 262 lower than that of the participants who successfully received the reminders.” These statements are more of results and findings, rather than conclusions of the study. It is important to make a transition from results/findings to conclusions in an organised and systematic manner. The conclusion should contain direct impressions from the results and findings. Conclusions are statements derived as logical extensions of the evidence produced by the study. It is not a rehearse of the results and findings but is informed by the results and findings. Rather, conclusions serve to reinforce or wrap-up the results and findings and helps to ease interpretation of the results and findings by summarising the giving some kind of summary deduction from the results and findings.
I see this study mainly focuses on a local issue analysis. Whether the method used in this study can be applied to other regions/countries. The conclusion should expand this aspect to attract international readers.
It will also be useful to highlight the contributions of the study and provide clear insights regarding policy implications and recommendations.
Author Response
Reviewer 1
- While the paper describes the sampling method at length, crucial details are missing. For example, how and why were the participants randomly assigned to three groups each month: the "no reminder” group (n = 458), the "letter reminder” group (n = 459), or the "telephone reminder” group (n = 460)?
Response: We appreciate your kind review and comments. As the reviewer pointed out, detailed information was missing in the previous version of the manuscript. In this revision, we have added some explanations in the methods section (lines 94-95 and 97-99).
“This study adopted a random sampling method to enhance the internal validity of the findings.” (lines 94-95)
“The staff provided a unique number to every participant and assigned them randomly to the three groups using a random number generator. This process was performed each month.” (lines 97-99)
- How was the survey ‘administered’?
Response: The random assignment and intervention were carried out by the staff of Yokohama City. I have added the explanations in the methods section (lines 97 and 109-110). The data (i.e., the outcome and variables on participants’ characteristics) was collected via the Data Management System of Yokohama City (this was already explained in the text.).
“Random assignment was conducted by the staff of Yokohama City.” (line 97)
“We adopted letter and telephone reminder interventions in this study. The interventions were administered by the staff of Yokohama City.” (lines 109-110)
- What are the assumptions for use of the chi-square test, Fisher's exact test, Kruskal-Wallis test, Bonferroni correction and one-way analysis of variance? Was the data set tested to assess its compliance with these assumptions?
Response: In accordance with your comment, we checked the normality of the distribution of the continuous variable using the Shapiro-Wilk test and found that they were not normally distributed (we have added the explanation; lines 160-162). Therefore, in the revised manuscript, we have used the Kruskal-Wallis test (non-parametric test) instead of one-way analysis of variance.
“For continuous variables, confirmed to be not normally distributed by the Shapiro-Wilk test, the non-parametric test was conducted (i.e., the Kruskal-Wallis test).” (lines 160-162)
- How did you handle the issue of hidden and overt bias?
Response: The design of this study was a random-controlled design, and therefore we expect that most bias (regarding internal validity, in particular) could be removed. However, there might remain hidden biases such as in the sampling process. For example, the sample of this study included those who were at risk of metabolic syndrome. Therefore, our results will not be generalizable to other populations like the healthy population. We have mentioned this point in the limitation section (lines 261-264).
“First, as previously mentioned, the current study targeted only those with metabolic syndrome or subjects at risk of developing this entity. The effects on other populations need to be investigated in future studies to enhance the external validity.” (lines 261-264)
- Can you attribute the changes recorded in the outcome to the telephone and/or letter reminder? Of course there are other factors (hidden and unhidden) that could course the changes. How did you handle these hidden and unhidden factors methodologically and econometrically?
Response: As mentioned above, the random controlled trial cannot enhance the external validity; but it can increase the interval validity. The random controlled design could control for potential confounders including hidden and unhidden. Therefore, we concluded that the results were caused by the difference in the reminding method.
- The discussion of the results needs to be improved. Currently, it is unfocused. In addition, much of the discussion is reiterating the results, while there is only limited discussion of the findings in light of previous and related literature. The discussion section overall would be improved by a much clearer focus on elucidating key findings in light of existing literature and how this study builds on/relates to past findings. Specifically, there needs to be significantly more references and linkage to relevant literature to back up many of the points made in this section.
Response: According to your advice, we have reviewed the discussion section. As the reviewer indicated, there have been some parts that reiterated the results. In the revised manuscript, we have modified the section by referring to some new papers.
- This conclusion is poorly written and needs to be reworked. Look at this excerpt from the section – “There was no difference in the utilization rate for either letter or telephone reminders compared with no reminders. However, in the telephone reminder group, 33.0% of the participants could not be reached through phone calls, and their utilization rate was 262 lower than that of the participants who successfully received the reminders.” These statements are more of results and findings, rather than conclusions of the study. It is important to make a transition from results/findings to conclusions in an organised and systematic manner. The conclusion should contain direct impressions from the results and findings. Conclusions are statements derived as logical extensions of the evidence produced by the study. It is not a rehearse of the results and findings but is informed by the results and findings. Rather, conclusions serve to reinforce or wrap-up the results and findings and helps to ease interpretation of the results and findings by summarising the giving some kind of summary deduction from the results and findings.
Response: According to your advice, we have also reviewed and modified the conclusion section. We have avoided rephrasing the results.
- I see this study mainly focuses on a local issue analysis. Whether the method used in this study can be applied to other regions/countries. The conclusion should expand this aspect to attract international readers.
Response: According to your advice, we have added this point in the conclusion section. Moreover, the generalizability of the findings has been described in the limitation section (lines 261-264). These two explanations can aid the understanding for international readers.
“First, as previously mentioned, the current study targeted only those with metabolic syndrome or subjects at risk of developing this entity. The effects on other populations need to be investigated in future studies to enhance the external validity.” (lines 261-264)
- It will also be useful to highlight the contributions of the study and provide clear insights regarding policy implications and recommendations.
Response: In the conclusion section of the revised manuscript, we have added sentences regarding the policy implications (lines 284-288).
“The results suggest setting a low priority of reminding people in the at-risk-for metabolic syndrome population. Nonetheless, this study possibly underestimated the effectiveness of reminders. Reminders using either letters or telephone calls are labor and cost incentive to some degree. Thus, more effective and efficient methods should be explored for implementing reminders.” (lines 284-288)

Reviewer 2 Report
I would like to thank Murayama et al. for the opportunity to review the manuscript of the article "Evaluating the effectiveness of letter and telephone reminders in promoting the use of specific health guidance in an at-risk-population with metabolic syndrome in Japan: A randomized controlled trial". In this article, the authors studied 2 options for reminders of specific health recommendations in this category of people. Surprisingly, the authors found that none of the methods affected the rates of use of specific health advice in the risk group for metabolic syndrome.
When reviewing the manuscript, I had the following questions and comments, to which I would like to receive answers from the authors:
1. I would like to clarify why only 2 reminder options were tested (letter and phone calls). Why didn't you use other means of communication with the patient (for example, SMS messages or e-mails [1])?
2. I would also like to clarify what were the specific recommendations for people with metabolic syndrome or the risk of developing it? In my opinion, the description in the text of the manuscript ("explanation of the specific health guidance (including information that the specific health guidance was free of charge)," "the expiration date of the specific health guidance," "information on the medical centers/hospitals /clinics where the specific health guidance is provided," and "telephone number for inquiries.") are extremely scanty and general in nature. Such a description of recommendations can be suitable for any medical situation.
3. I think it should be clarified whether the recommendations were of a standard nature or were they personalized [2]?
4. As far as I understand, everyone included in the study received health checkups. At the same time, out of 13,638 people who were eligible for specific health guidance based on national criteria, 10,763 people required immediate medical attention. The question arises - is it possible for that relatively small group with a metabolic syndrome or a risk of its development (1377 people) to also issue recommendations immediately after a medical examination, and not try to convey information through letters or phone calls?
5. The manuscript was able to trace the number of those who did not answer the phone. However, it is possible that some of the letters were not received or read, which could affect the frequency of the utilization rates of specific health guidance in the “letter reminder” group. In my opinion, this is a limitation of the study and should be included in the Study Limitations section.
Minor:
In table 2, when comparing the three groups, no statistical difference was noted (p = 0.093), so the subsequent post-hoc test is meaningless. Therefore, the last column in table 2 is redundant and should be deleted.
References:
1. Starr, K., McPherson, G., Forrest, M., & Cotton, S. C. (2015). SMS text pre-notification and delivery of reminder e-mails to increase response rates to postal questionnaires in the SUSPEND trial: a factorial design, randomised controlled trial. Trials, 16, 295. https://doi.org/10.1186/s13063-015-0808-9
2. Gidlow, C. J., Ellis, N. J., Riley, V., Chadborn, T., Bunten, A., Iqbal, Z., Ahmed, A., Fisher, A., Sugden, D., & Clark-Carter, D. (2019). Randomised controlled trial comparing uptake of NHS Health Check in response to standard letters, risk-personalised letters and telephone invitations. BMC public health, 19(1), 224. https://doi.org/10.1186/s12889-019-6540-8
Author Response
Reviewer 2
- I would like to clarify why only 2 reminder options were tested (letter and phone calls). Why didn't you use other means of communication with the patient (for example, SMS messages or e-mails [1])?
Response: We appreciate your kind review and comments. We agree with your comment. This was the first trial to examine the effect of a reminder. Therefore, we have adopted letter and phone reminders that were easy to implement. As a next step, we are currently considering a reminder using SMS. We have added the explanation on this in the limitation section (lines 277-279) and the conclusion section (lines 294-298).
“Finally, this study investigated the effectiveness of letter and telephone reminders. However, there are some other reminder options (e.g., short message service [SMS], e-mail), and these effects should be examined in the future.” (lines 277-279)
“In addition, there are other methods such as SMS and e-mail rather than the letter and telephone reminder methods investigated in this study, and more effective methods will probably emerge with the advancement of technology. Such methods also need to be verified using a robust design.” (lines 294-298)
- I would also like to clarify what were the specific recommendations for people with metabolic syndrome or the risk of developing it? In my opinion, the description in the text of the manuscript ("explanation of the specific health guidance (including information that the specific health guidance was free of charge)," "the expiration date of the specific health guidance," "information on the medical centers/hospitals /clinics where the specific health guidance is provided," and "telephone number for inquiries.") are extremely scanty and general in nature. Such a description of recommendations can be suitable for any medical situation. I think it should be clarified whether the recommendations were of a standard nature or were they personalized [2]?
Response: As the reviewer indicated, the contents of the recommendation were general in nature. The main point of the study was the way of reminding. For example, the letter reminder and telephone reminder included almost the same information, and the information was not personalized (we have added this point in the methods section; lines 110-111); the difference was mainly the way the reminder was given.
“The information provided to the participants via either letter or telephone call was not personalized.” (lines 110-111)
- As far as I understand, everyone included in the study received health checkups. At the same time, out of 13,638 people who were eligible for specific health guidance based on national criteria, 10,763 people required immediate medical attention. The question arises - is it possible for that relatively small group with a metabolic syndrome or a risk of its development (1377 people) to also issue recommendations immediately after a medical examination, and not try to convey information through letters or phone calls?
Response: The criteria to require immediate medical attention was defined by the Ministry of Health, Welfare, and Labour in Japan. Based on the criteria, health insurers have to recommend them to consult a doctor without reminding the use of specific health guidance. The basic procedure by the national guideline for specific health examination/guidance was defined as ‘to remind to receive the specific health guidance when the person was judged as at-risk of metabolic syndrome’ and not ‘requiring immediate medical consultation’. Therefore, the recommendation for immediate medical consultation will not be made for those at-risk of metabolic syndrome.
- The manuscript was able to trace the number of those who did not answer the phone. However, it is possible that some of the letters were not received or read, which could affect the frequency of the utilization rates of specific health guidance in the “letter reminder” group. In my opinion, this is a limitation of the study and should be included in the Study Limitations section.
Response: We agree with your opinion. We could not investigate how many participants actually read the letter reminder. We have added this point in the limitation section (lines 273-277).
“Fifth, we were not able to investigate how many participants in the letter reminder group actually read the letter. The effect of the letter reminder might differ between those who read the letter and those who did not. This means that we might have underestimated the effect of the letter.” (lines 273-277)
- In table 2, when comparing the three groups, no statistical difference was noted (p = 0.093), so the subsequent post-hoc test is meaningless. Therefore, the last column in table 2 is redundant and should be deleted.
Response: We have understood your comment. However, multiple comparisons had another meaning which was to examine the difference in the outcome between the two groups, and that was our interest. Therefore, in the revision, we have retained the multiple comparisons in the table. The term “post-hoc” was not correct, and thus we removed this from the manuscript.

Round 2
Reviewer 1 Report
Corrections effected
Reviewer 2 Report
I am quite satisfied with the answers to my questions, I have no other comments.